# HIV, HCV and HIV-HCV Coinfections in the General Population versus Inmates from Romania

**DOI:** 10.3390/v16081279

**Published:** 2024-08-10

**Authors:** Camelia Sultana, Carmine Falanga, Grațiana Chicin, Laurențiu Ion, Camelia Grancea, Daniela Chiriac, Adriana Iliescu, Andrea Gori

**Affiliations:** 1Faculty of Medicine, “Carol Davila” University of Medicine and Pharmacy, 050474 Bucharest, Romania; 2“Ștefan S. Nicolau” Institute of Virology, 030304 Bucharest, Romania; cgrancea@yahoo.co.uk (C.G.); dana.chiriac1976@gmail.com (D.C.); 3ANLAIDS, Sezione Lombarda, 20154 Milan, Italy; carmine.falanga@anlaidslombardia.it; 4General Medicine Department, “Vasile Goldiș” Western University of Arad, 310045 Arad, Romania; gchicin@gmail.com; 5National Institute of Public Health, 050463 Bucharest, Romania; 6Infectious Diseases Department, “Victor Babeș” Clinical Hospital of Infectious and Tropical Diseases, 030303 Bucharest, Romania; 7Jilava Penitentiary Hospital, 759712 Bucharest, Romania; adriana.iliescu@anp.gov.ro; 8Department of Biomedical and Clinical Sciences “L. Sacco”, University of Milan, 20122 Milan, Italy; andrea.gori@uimi.it

**Keywords:** HIV, HCV, HIV-HCV coinfection, micro-elimination, inmates, IDUs, STDs, strong association

## Abstract

The objective of this study was to analyze the epidemiological links of the human immunodeficiency virus (HIV), hepatitis C virus (HCV) and HIV-HCV coinfections to less studied types of transmission in certain populations. We performed an observational, prospective study on 903 patients aged between 15–87 years who took part in the Open Test Project. They were divided in two subgroups: general population vs. individuals from prisons who were questioned about multiple risk factors. A chi-square independence test was used to establish correlations between risk factors and results of screening tests. Logistic regression was used to calculate the probability of a reactive screening test based on each independent risk factor and age. HIV was very strongly associated with unprotected sexual intercourse with HIV-positive partners (the strongest association), unprotected sexual intercourse with sex workers, newly diagnosed sexually transmitted diseases (STDs), intravenous drug users (IDUs) and sharing injecting materials. In the case of HCV reactive tests, very strong associations have been established with IDUs (the strongest association), unprotected sex with IDUs and sharing injecting materials. Our study indicates the need for implementing targeted public health programs, tailored to the local epidemiology that can ultimately lead to micro-elimination of hepatitis and HIV infections in this area.

## 1. Introduction

The World Health Organization’s sustainable development goals for 2030 aim to end the epidemic of acquired immunodeficiency syndrome (AIDS), combat hepatitis and other communicable and sexually transmitted diseases (STDs) by 2030 [1]. Still, a more realistic objective may be the micro-elimination of human immunodeficiency virus (HIV) and hepatitis C virus (HCV) from population segments with targeted strategies of prevention and treatment. The basic idea is that micro-elimination will eventually lead to macro-elimination.

Moreover, infection with HIV and HCV is frequently encountered in individuals with similar risk factors and shared parenteral transmission routes. HIV poses a major global health challenge, worsened by rising emigrant numbers [2]. There are currently 18,015 people living with HIV in Romania, a significant proportion of them being long-term survivors from the 1990 cohort, parenterally infected during childhood [3,4]. However, only 13,759 receive effective treatment or post-exposure prophylaxis [3]. Therefore, more effort should be put into prompt and early identification and management of new cases.

On the other hand, HCV affects about 170 million people worldwide, significantly outnumbering HIV-1 infections [5]. Most HCV acute cases develop chronic hepatitis, leading to severe complications such as fibrosis, cirrhosis or hepatocellular carcinoma without diagnosis and treatment [6]. Rapid and accurate diagnosis is the key to preventing the spread of HCV infections, Romania being the country with the highest prevalence of HCV infections among the Balkan countries [7]. While serological screening and risk factor identification can reduce the risk of parenterally transmitted diseases, new cases of HCV infection still occur through dangerous practices.

The present study aims to analyze HIV and HCV infections transmission pathways in Romania and the epidemiological links to the main novel transmission types. The study assessed the risk of a positive screening test when exposed to an extended list of predefined risk factors for HIV and HCV. Assuming that individual exposure to risk factors varies, this study seeks to enhance understanding of HIV and HCV transmission dynamics for improved personalized diagnosis and treatment guidelines. Nonetheless, the decisive goal is to accurately identify up-to-date risk factors, improve screening and promote voluntary testing.

## 2. Materials and Methods

We performed an observational, population oriented, prospective study on 903 individuals with ages between 13–87 years who participated in the European Open Test Project, carried out in Stefan S. Nicolau Institute of Virology from Bucharest and Profilaxia Medical Centre from Timișoara, Romania, coordinated by ANLAIDS (Associazione Nazionale per la Lotta contro l’AIDS—Sezione Lombarda), Italy, with funding from Otto per Mille della Chiesa Valdese, Italy. The participants were randomly chosen; all the people who accepted to be tested and be part of the project were included in the study and tested free of charge for HIV and HCV infections.

A questionnaire was used to collect information from the study population during November 2019 and February 2023. All the participants signed an informed consent and received medical counselling before and after testing. The protocol was approved by the Ethics Commission of the Institute of Virology. Our research was conducted within the Declaration of Helsinki guidelines and under the terms of all local legislation. Patients were questioned about the following risk factors: sexual intercourse with men, women or both, abstinence, condom use during sexual intercourse as a means of STD prevention, offering sexual services in exchange of drugs and money, newly diagnosed STDs, incarceration, unprotected sexual intercourse with sex workers, intravenous drug user (IDU) status, sexual intercourse with IDUs, HIV positive people and men who have sex with men (MSM) and sharing needles, syringes, spoons, filters or other injecting materials, all in the previous 12 months. Patients were also evaluated for demographic factors (gender, date of birth, age, nationality, residence place—urban or rural), and previous HIV and HCV testing.

Rapid qualitative tests were used to detect antibodies for screening of HIV and HCV from blood or saliva (OraQuick HIV and HCV tests); briefly, after collecting samples from the oral cavity or peripheral blood, the samples were processed in a developer vial. The presence of antibodies in the sample is indicated by the appearance of a reddish-purple line in both the control zone and test zone of the device (reactive test). All the reactive tests were confirmed using an immunoenzymatic test.

Patients’ data were processed using the JASP statistics program; to establish correlations between the risk factors and the result of the screening tests, we used a chi-square independence test. The degree of association was evaluated with the φ (phi) association coefficient (for binary variables) and Cramer’s V coefficient (for nonbinary variables). We set a *p* value at 0.05 (confidence interval, CI = 95%). The degree of freedom (df value) is obtained from the contingency table. Frequencies lower than 5 needed a Yates continuity correction. Coefficients > 0.25 indicated a very strong association between the risk factor and the result of the screening test; 0.15–0.25—strong association; 0.1–0.15—moderate association; 0.05–0.1—weak association; 0–0.05—very weak or absent association.

Furthermore, to calculate the probability of a reactive screening test based on each independent risk factor and age, logistic regression was used. Results are presented as odds ratio (OR). OR > 1 is associated with a higher odds of the intended outcome, OR = 1 indicates no associations, while OR < 1 predicts lower odds.

## 3. Results

A total of 903 Romanian individuals with different risk profiles were introduced in the study and subsequently divided into two subgroups: general population (806 persons) and individuals from prisons (97 persons). The association between individual risk factors and the screening test results for HIV, HCV and HIV-HCV coinfection were compared between the two subgroups.

### 3.1. General Characteristics of the Patients

The general population cohort, with a mean age of 39.37 ± 14.54 years, consisted mainly of women (63.89%) from urban areas (Bucharest—38.33, Timișoara—28.16%). A total of 26 individuals had reactive HIV screening tests (3.22%), 22 had reactive HCV screening tests (2.72%) and seven had reactive tests for both HIV and HCV (0.86%). The mean age for HIV reactive individuals was 36.65 ± 8.93 years, while for HCV reactive population it was 43.31 ± 11.91 years, older than that of the HIV group.

Prison inmates averaged 41.66 ± 9.28 years, were mostly men (92.78%) from Jilava Penitentiary Hospital. A total of 55 had reactive HIV tests (56.7%), 49 had reactive HCV tests (50.51%) and 30 had HIV-HCV coinfections (30.92%). The mean age for HIV reactive inmates was 41.12 ± 8.72 years and the mean age in HCV reactive prisoners was 39.3 ± 6.78 years. The general characteristics of the study groups are presented in Table 1.

In total, 3.22% from the general population, as compared to 56.7% from the prisoners had reactive HIV screening tests; 2.73% from the people in the general population and 50.51% from the imprisoned had reactive HCV screening tests, while 0.86% from the general population and 30.92% from the prisoners had reactive tests for both HIV and HCV.

### 3.2. Correlations between the Risk Factors and the Outcomes of the Screening

We assessed how heterosexual intercourse within the last 12 months with women influences screening results in both populations. A strong association between the risk factor and the HIV screening test result was observed only in correctional facilities populations. Other associations are detailed in Table 2, Table 3, Table 4 and Table 5, which also includes comparative data regarding the odds for positive HIV and HCV tests simultaneously or separately, alongside the chi-square test and logistic regression analysis results.

Heterosexual intercourse with men in the previous 12 months was evaluated as a potential risk factor as well. A very strong correlation (φ = 0.32, *p* < 0.05) was observed for HIV among individuals in prisons. A strong association was found for HCV and HIV-HCV coinfection (φ = 0.2, *p* = 0.1 and φ = 0.22, *p* = 0.06, respectively). Within the general population, HIV showed a moderate association. HCV and coinfection are presented in Table 2, Table 3, Table 4 and Table 5, along with the relationships for sexual activity with both men and women in the last 12 months. All available results are for exposures to risk factors over the past 12 months.

The correlation between abstinence from sexual intercourse in the last 12 months and the screening tests results for HIV and HCV in the prison population was very strong (Cramer’s V coefficient = 0.28, *p* = 0.01 and 0.3, *p* = 0.01, respectively) and strong in the case of coinfection (Cramer’s V coefficient = 0.23, *p* = 0.07). Abstinence reduced the chances of reactive HIV and HCV tests in inmates, especially among young patients.

When evaluating unprotected sexual intercourse with penetration within the last 12 months as a risk factor, it was observed that 451 individuals from the general population and 69 individuals from prisons engaged in this risky behavior, while eight individuals from the second population were unable to recall exposure. Only 5.32% of individuals from the exposed general population presented for an HIV reactive screening test, while 57.97% of exposed individuals from the prison population had reactive HIV screening tests. A total of 50% of individuals who reported not remembering exposure also had reactive HIV screening tests. In the case of HCV reactive tests, a strong association was established (Cramer’s V coefficient = 0.23, *p* = 0.07) among individuals in prisons. Other correlations are detailed in Table 2, Table 3, Table 4 and Table 5.

The participants were asked whether they had been involved in sexual services in exchange for drugs or money in the last 12 months. In the general population, we noticed a prevalence of 42.85% of reactive HIV tests and 71.42% of reactive HCV tests among individuals exposed to this risk factor. The identified association was strong (φ = 0.21, *p* < 0.001) for HIV and very strong for HCV and coinfection (φ = 0.39, *p* < 0.001 and φ = 0.42, *p* < 0.001, respectively). Among individuals in prisons, only the associations with HIV and coinfection were strong (φ = 0.2, *p* = 0.1 and φ = 0.17, *p* = 0.21). The correlations between unprotected penetrating sexual intercourse with sex workers are presented in Table 2, Table 3, Table 4 and Table 5.

Individuals diagnosed with STDs in the last 12 months were much more likely to have reactive HIV (OR = 2.77, *p* = 0.3) and HCV (OR = 1.95, *p* = 0.52) screening tests among individuals in prisons. Newly diagnosed STDs were strongly associated with reactive HIV screening tests in both populations (general population: Cramer’s V coefficient = 0.33, *p* < 0.001, prison: φ = 0.3, *p* < 0.05). Additionally, in the general population, the odds of obtaining a positive screening test for exposed individuals compared to the odds of obtaining a positive one for individuals not recalling exposure were higher for HIV (OR = 7.5, *p* = 0.03). The association with HCV was strong in both populations (general population: Cramer’s V coefficient = 0.19, *p* < 0.001, prison: φ = 0.24, *p* = 0.02).

The association between unprotected penetrating sexual intercourse with HIV+ individuals in the last 12 months and the results of HIV screening tests was one of the strongest in both groups. In the general population, 60% of exposed individuals presented with a reactive test, while only 2.52% of unexposed individuals did. Among individuals in prisons, 92.3% of those who did not recall their exposure were presented with a reactive HIV screening test. Thus, very strong correlations (Cramer’s V coefficient = 0.36, *p* < 0.001) were established in the general population, as well as among individuals in prisons (Cramer’s V coefficient = 0.33, *p* < 0.05).

In the case of HCV, the identified correlation was strong (Cramer’s V coefficient = 0.2, *p* = 0.13) only among individuals in prisons. In the case of HIV-HCV coinfections, a very strong association was also established among individuals in prisons (Cramer’s V coefficient = 0.28, *p* = 0.02).

A strong association (Cramer’s V coefficient = 0.15, *p* < 0.001 in the general population and φ = 0.16, *p* = 0.1 among individuals in prisons) was observed between sexual intercourse with men who have sex with men (MSM) and the results of the HIV screening tests in both tested populations. In general population, 25% of exposed individuals subsequently had a reactive HIV test. Other correlations are presented in Table 2, Table 3, Table 4 and Table 5.

In the general population, intravenous drug users showed one of the strongest associations with screening tests for HIV, HCV and HIV-HCV coinfection (HIV: Cramer’s V coefficient = 0.32, *p* < 0.001, HCV: Cramer’s V coefficient = 0.59, *p* < 0.001, coinfection: Cramer’s V coefficient = 0.57, *p* < 0.001). On the other hand, in the prison population, a very strong association was only established for HCV (φ = 0.46, *p* < 0.05). The association with the coinfection was strong (φ = 0.17, *p* = 0.09). Comparative data for the two populations are presented in Figure 1.

Within the same risk group, behaviors associated with intravenous drug use, such as sharing needles, syringes, spoons, filters and other equipment in the past 12 months, were also evaluated. For needles and syringes, among the general population, a very strong association was established between the risk factor and the screening test results (HIV: Cramer’s V coefficient = 0.32, *p* < 0.001; HCV: 0.42, *p* < 0.001; coinfection: 0.54, *p* < 0.001).

In the case of individuals in correctional facilities, a very strong association was only established for HCV (Cramer’s V coefficient = 0.27, *p* = 0.02). For HIV and coinfection, strong associations were identified (Cramer’s V coefficient = 0.21, *p* = 0.11; 0.21, *p* = 0.1, respectively). In total, 68.42% of those who shared needles or syringes had reactive HCV screening tests. Comparisons regarding sharing needles and syringes between the two populations are presented in Figure 2.

Likewise, 17.02% of individuals in the general population who shared the second category of objects had reactive HIV tests (Table 2, Table 3, Table 4 and Table 5).

Unprotected sexual intercourse with intravenous drug users in the past 12 months was evaluated as a potential risk factor. The results of its association with the screening test results are presented in Figure 3. In total, 30.76% of individuals in the general population and 54.54% in correctional facilities who adopted this behavioral pattern had reactive HIV tests.

In the general population, this risk factor was strongly associated with HCV (Cramer’s V coefficient = 0.41, *p* < 0.001) and HIV-HCV coinfection (Cramer’s V coefficient = 0.41, *p* < 0.001), and moderately associated with HIV (Cramer’s V coefficient = 0.2, *p* < 0.001). In contrast, among incarcerated individuals, the association was very strong only in the case of HCV (Cramer’s V coefficient = 0.37, *p* < 0.05).

HIV reactive tests in the general population was strongly associated with unprotected sex with HIV-positive partners (strongest association), sex workers, new STDs, IDUs and sharing injecting materials (syringes/needles). Among inmates, the strongest association was also unprotected sexual intercourse with HIV+ partners, followed by new STDs. Very strong connections were also established with heterosexual intercourse with men and abstinence in the prisons, while these connections were weak in the general population.

HCV reactive tests showed very strong associations with intravenous drug use (strongest association in both populations), unprotected sex with IDUs and sharing injecting materials (syringes/needles/spoons/filters/water) in both populations and very strong associations with trading sexual services in the general population. In inmates, abstinence was very strongly associated with HCV screening tests, while this association was weak in the general population. HIV-HCV coinfection patterns mirrored HCVs in the general population, but among inmates, it was very strongly associated with unprotected sexual intercourse with HIV-positive partners. A general overview of the association between HIV, HCV and HIV-HCV reactivity and various reported risk factors is presented in Figure 4.

## 4. Discussion

Our study reports an extremely strong association between intravenous drug use and the results of the HIV, HCV and HIV-HCV coinfection screening tests in the general population. Since 2011, Romania has faced an HIV epidemic among intravenous drug users, reported in patients exposed to heroin abuse who recently started using psychostimulant drugs too. One third of intravenous drug users were then reported with HIV infection while in detention [8]. In 2015, Romania launched two programs aimed to improve prevention and medical and social care for drug users [9]. These programs also targeted the inmates from Jilava Penitentiary in Bucharest, a population included in this study as well. Prisoners had significantly higher rates of positive screening tests. However, the 2020 report from Romania’s National Anti-Drug Agency highlighted a decrease in newly diagnosed HIV infections among IDUs from 32.2% in 2013 to 11% in 2019 [10].

Evidence-based interventions should be integrated and brought to scale, while upcoming programs should focus on the underlying structural drivers of HIV transmission [11]. Regarding the need to put an end to outbreaks in these risk groups, other issues such as homelessness and poverty should also be addressed in an integrative manner so that not only IDUs are spared of an endless cycle of HIV transmission, but also their communities.

Around 10 million people are incarcerated or in pretrial detention worldwide, with the United States, China and Russia accounting for half of them. One in four prisoners has HCV infection [12]. Incarceration increases the risk of HIV and HCV due to associated risky behaviors like unprotected sex, tattooing, intravenous drug use, sexual violence and sharing injection equipment [13]. The risk is amplified by the scarcity of treatment services in prisons, as only 28 countries worldwide provide condoms in prisons [12].

Sexual partners of IDUs face a significant infection risk, which increases with multiple partners [14]. We identified a strong association in the general population. Though underreporting may exist, similar data are also reported in the case of HCV [15,16]. In our study, a very strong association with HCV was identified in both groups. Furthermore, the association between intravenous drug use and HCV is well-known and the risk is even higher when sharing needles and injecting tools [17]. This can be attributed to the common behavioral patterns of intravenous drug users.

Therefore, despite the decreasing trends of HIV and HCV infections incidence rates in Romania, the risk of HIV and HCV transmission persists and requires further studies, especially considering the dynamic changes in drug consumption behaviors in favor of new psychoactive substances which comprised 6.3% of all drugs used in Romania in 2019 [10].

Unprotected sex with known HIV-positive individuals in the past year was a major risk factor for HIV infection in our study, showing strong associations with HIV reactive tests and even stronger with HCV and coinfection, particularly among prisoners. Interestingly, inmates unaware of their exposure still had a high positivity rate in screening tests, with 92.3% of tested prisoners being positive.

Unprotected homosexual intercourse significantly raises HIV infection risk, anal sexual intercourse posing the highest risk [15]. Our study identified a strong association with HIV, but not HCV in both populations, which is consistent with other reported studies, reporting its uncommon transmission through unprotected sexual intercourse. A recent study from Mexico suggested an association between drug use during sexual activities (“chemsex”) and HCV transmission, especially following the use of ethyl chloride [18]. Following this pattern, the data from the current study should be extended, considering the lack of information regarding additional sexual practices of the individuals in this study.

The current study indicates a strong association between heterosexual activity with women and HIV screening test results in prisons and a moderate association in the general population. In the case of intercourse with men, the prison group had a higher transmission risk with very strong associations for HIV and strong for HCV and HIV-HCV coinfection. Research on 563 heterosexual couples across nine European countries indicated higher transmission rates from men to women with 12% of men and 20% of women contracting HIV from their opposite-sex partners [19].

Individuals who engaged in commercial sex show a high prevalence of STDs: HIV, chlamydia, gonorrhea, syphilis, including HCV, likely due to multiple risk factors [20]. Our study showed a very strong link between unprotected sexual activity with sex workers and HIV test results in the general population, but a weak association in prisons and with HCV both groups, similar with a study from Zimbabwe, attributing this result to the multiplicity of risk factors or underreporting of this behavior [21]. Likewise, the last HIV/AIDS surveillance report in Europe from 2023 indicate that the new cases of HIV in Eastern Europe are associated mostly with heterosexual transmission (men 38.3%, women 33.1%) [22]. Our study found STD history to be a potent risk factor for HIV and HCV, with meta-analyses supporting this association, particularly emphasizing population variability and highlighting the possibility of association between various past non-viral STDs and HIV acquisition [23,24,25]. Moreover, other coinfections also influence the risk of HIV acquisition as well as the course of the disease; the Epstein–Barr virus (EBV) coinfection lowers gamma-interferon (IGN-γ) levels which leads to important pathogenic consequences [26]. However, analyzing these results is challenging because of the possible overlapping sexual behaviors and other risk factors [24].

Our study’s limitation lies in the cohort composition, acknowledging the fact that the percentages of males and females in the two populations are different, however, the National Administration of Penitentiaries’ report from 31 January 2023 stated that there were 22,030 men (95.62%) and 1010 women (4.38%) imprisoned in Romania in incarceration units from their administration at the end of 2022. Given this high proportion of imprisoned men, the 92.78% of male gender reported in our study is representative for the inmate’s population [25].

Besides the limited number of MSM participants from both populations, our study also lacks a sufficient number of participants which hinders to some degree broad generalization. However, to our knowledge, this is the first study to compare the risk factors in incarcerated people with the ones from the general population in Romania, and, although a decreasing trend in HIV and HCV incidence rates are reported in this country, the risk of transmission persists, requiring innovative studies to achieve the micro-elimination of the two infections in this area.

## 5. Conclusions

Our study underscores the importance of local, targeted and comprehensive public health programs to enhance HIV and HCV diagnosis and management. Key findings include a very strong association between unprotected sexual intercourse with HIV positive individuals and positive HIV tests in both populations. Trading sex for money or drugs had a strong HIV association in both populations, but for HCV this was very strong in the general population, but weak among inmates. Sharing injection equipment was highly associated with HIV, HCV and coinfection in both groups. Unprotected sex with IDUs was very strongly associated with HCV in both populations, and with HIV in the general population but weakly in the inmate’s population. Our study indicates the need for implementing targeted public health programs tailored to the local epidemiology that can ultimately lead to micro-elimination of hepatitis and HIV infections in this area.

## Figures and Tables

**Figure 1 viruses-16-01279-f001:**
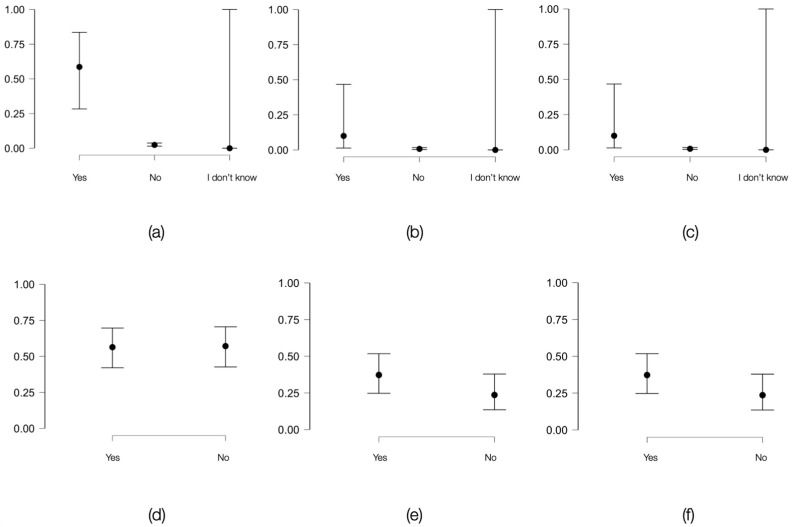
Variation in the probability of obtaining positive HIV and HCV screening tests or the presence of HIV-HCV coinfection based on the presence of the risk factor (intravenous drug users) among individuals in the general population and in the inmate population: (**a**) graphical representation of how the probability of a positive HIV screening test varies based on the presence or absence of the risk factor among individuals of the same age in the general population; (**b**) HCV in the general population; (**c**) HIV-HCV coinfection in the general population; (**d**) graphical representation of how the probability of a positive HIV screening test varies based on the presence or absence of the risk factor among individuals of the same age in the inmate population; (**e**) HCV in the inmate population; (**f**) HIV-HCV coinfection in the inmate population.

**Figure 2 viruses-16-01279-f002:**
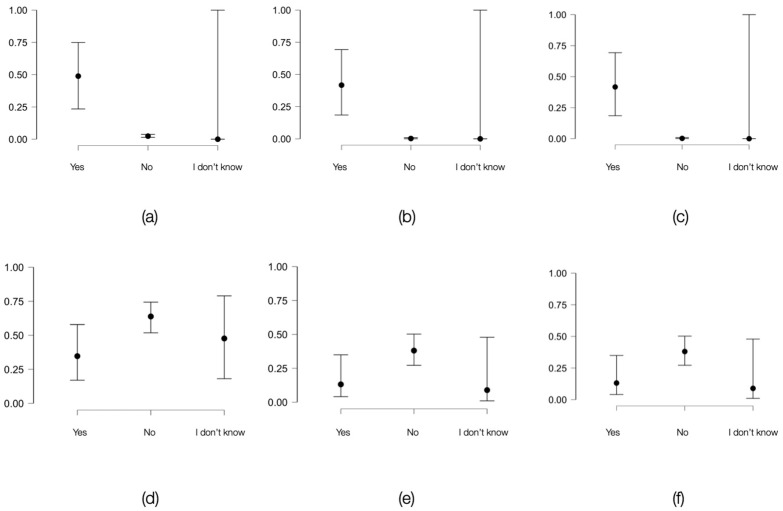
Variation in the probability of obtaining positive HIV and HCV screening tests or the presence of HIV-HCV coinfection based on the presence of the risk factor (sharing injection materials such as syringes and needles within the last 12 months) among individuals in the general population and in the inmate population: (**a**) graphical representation of how the probability of a positive HIV screening test varies based on the presence or absence of the risk factor among individuals of the same age in the general population; (**b**) HCV in the general population; (**c**) HIV-HCV coinfection in the general population; (**d**) graphical representation of how the probability of a positive HIV screening test varies based on the presence or absence of the risk factor among individuals of the same age in the inmate population; (**e**) HCV in the inmate population; (**f**) HIV-HCV coinfection in the inmate population.

**Figure 3 viruses-16-01279-f003:**
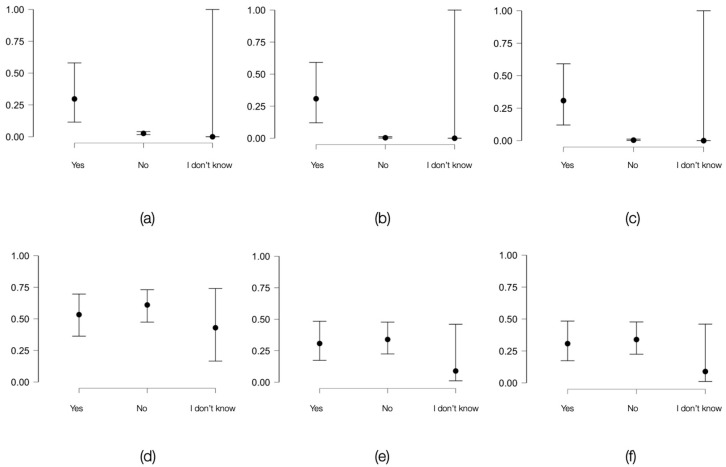
Variation in the probability of obtaining positive HIV and HCV screening tests or the presence of HIV-HCV coinfection based on the presence of the risk factor (unprotected sexual intercourse with intravenous drug users in the past 12 months) among individuals in the general population and in the inmate population: (**a**) graphical representation of how the probability of a positive HIV screening test varies based on the presence or absence of the risk factor among individuals of the same age in the general population; (**b**) HCV in the general population; (**c**) HIV-HCV coinfection in the general population; (**d**) graphical representation of how the probability of a positive HIV screening test varies based on the presence or absence of the risk factor among individuals of the same age in the inmate population; (**e**) HCV in the inmate population; (**f**) HIV-HCV coinfection in the inmate population.

**Figure 4 viruses-16-01279-f004:**
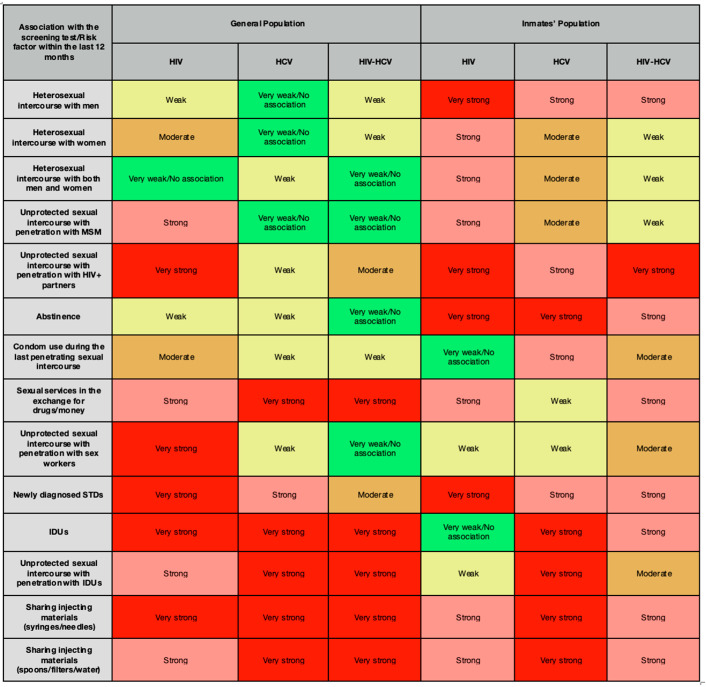
Established associations between risk factors and the results of the screening tests for HIV, HCV and coinfection in the general population and inmate population based on the phi coefficient and Cramer’s V coefficient.

**Table 1 viruses-16-01279-t001:** General characteristics of the patients.

Characteristics		General Population		Inmates		Total Population	
		Number	Percentage	Number	Percentage	Number	Percentage
Gender	Women	515	63.89%	7	7.22%	522	57.8%
	Men	291	36.11%	90	92.78%	381	42.2%
Mean Age (Years ± Standard Deviation)		39.37 ± 14.54	-	41.66 ± 9.28	-	39.61 ± 14.09	-
Nationality	Romanian	778	96.52%	97	100%	875	96.89%
	Other *	28	3.48%	0	0%	28	3.11%
Main Reasons for Testing	Assessing General Condition	518	64.26%	9	9.27%	527	58.36%
	Regular Check-up	64	7.94%	52	53.6%	116	12.84%
	Unprotected Vaginal Sex	34	4.21%	2	2.06%	36	3.98%
	Unprotected Oral Sex	32	3.97%	9	9.27%	41	4.54%
	Unprotected Anal Sex	16	1.98%	6	6.18%	22	2.43%
	Other **	142	17.61%	19	19.58%	161	17.82%

Legend: assessed characteristics of the general population, inmate’s population and total studied population presented in absolute values and percentages; * this includes Moldavian, Serbian, Lebanese and Turkish people; ** this includes screening before conception, recently diagnosed partner with HIV, abandonment of condom use, condom break during sexual intercourse, partner’s request and other unspecified reasons.

**Table 2 viruses-16-01279-t002:** The correlation between risk factors and the outcomes of HIV, HCV and HIV-HCV coinfection screening tests in the general population (chi-squared test).

Risk Factor		Chi-Squared Test		
		Number of Positive Tests among Exposed (Percentage of Positive Tests among Exposed)	Number of Positive Tests among Unexposed (Percentage of Positive Tests among Unexposed)	Adjusted *p* Value
Sexual intercourse with men	HIV	10 (2.1%)	16 (4.84%)	0.04
	HCV	11 (2.31%)	11 (3.33%)	0.51
	HIV-HCV	2 (0.42%)	5 (1.51%)	0.2
Sexual intercourse with women	HIV	15 (6.3%)	11 (1.93%)	3
	HCV	8 (3.36%)	14 (2.46%)	0.63
	HIV-HCV	4 (1.68%)	3 (0.52%)	0.23
Sexual intercourse with both men and women	HIV	0 (0%)	26 (3.26%)	0.85
	HCV	1 (12.5%)	21 (2.63%)	0.23
	HIV-HCV	0 (0%)	7 (0.87%)	0.96
Unprotected sexual intercourse with men who have sex with men (MSM)	HIV	3 (25%)	23 (2.9%)	<0.001
	HCV	0 (0%)	22 (2.78%)	0.8
	HIV-HCV	0 (0%)	7 (0.88%)	0.93
Unprotected sexual intercourse with HIV+ partners	HIV	6 (60%)	20 (2.52%)	<0.001
	HCV	1 (10%)	21 (2.65%)	0.34
	HIV-HCV	1 (10%)	6 (0.75%)	7
Abstinence	HIV	22 (4.43%)	4 (1.29%)	0.02
	HCV	18 (3.62%)	4 (1.29%)	0.07
	HIV-HCV	4 (0.8%)	3 (0.96%)	1
Condom use during the last penetrating sexual intercourse	HIV	2 (0.56%)	24 (5.32%)	<0.001
	HCV	5 (1.4%)	17 (3.76%)	0.06
	HIV-HCV	0 (0%)	7 (1.55%)	0.04
Sexual intercourse in exchange for drugs/money	HIV	3 (42.85%)	23 (2.87%)	<0.001
	HCV	5 (71.42%)	17 (2.12%)	<0.001
	HIV-HCV	3 (42.85%)	4 (0.5%)	<0.001
Unprotected sexual intercourse with sex workers	HIV	4 (50%)	22 (2.76%)	<0.001
	HCV	1 (12.5%)	21 (2.63%)	0.23
	HIV-HCV	0 (0%)	7 (0.87%)	0.96
Newly diagnosed sexually transmitted diseases (STDs)	HIV	2 (13.33%)	18 (2.31%)	<0.001
	HCV	2 (13.33%)	17 (2.18%)	<0.001
	HIV-HCV	0 (0%)	6 (0.77%)	0.01
Intravenous drug users (IDUs)	HIV	8 (38.09%)	18 (2.29%)	<0.001
	HCV	13 (61.9%)	9 (1.14%)	<0.001
	HIV-HCV	7 (33.33%)	0 (0%)	<0.001
Unprotected sexual intercourse with IDUs	HIV	4 (30.76%)	22 (2.78%)	<0.001
	HCV	7 (53.84%)	14 (1.77%)	<0.001
	HIV-HCV	4 (30.76%)	3 (0.38%)	<0.001
Common use of injecting materials (syringes/needles)	HIV	6 (50%)	20 (2.52%)	<0.001
	HCV	7 (58.33%)	15 (1.89%)	<0.001
	HIV-HCV	5 (41.66%)	2 (0.25%)	<0.001
Common use of injecting materials (spoons/filters/water)	HIV	8 (17.02%)	18 (2.39%)	<0.001
	HCV	9 (19.14%)	13 (1.72%)	<0.001
	HIV-HCV	7 (14.89%)	0 (0%)	<0.001

Legend: the results of the chi-squared test with the number and percentage of positive tests among exposed and unexposed persons in the general population; *p* < 0.05 was considered significant.

**Table 3 viruses-16-01279-t003:** The correlation between risk factors and the outcomes of HIV, HCV and HIV-HCV coinfection screening tests in the inmate population (chi-squared test).

Risk Factor		Chi-Squared Test		
		Number of Positive Tests among Exposed (Percentage of Positive Tests among Exposed)	Number of Positive Tests among Unexposed (Percentage of Positive Tests among Unexposed)	Adjusted *p* Value
Sexual intercourse with men	HIV	1 (10%)	54 (62.06%)	<0.05
	HCV	8 (80%)	41 (47.12%)	0.1
	HIV-HCV	0 (0%)	30 (34.48%)	0.06
Sexual intercourse with women	HIV	52 (59.77%)	3 (30%)	0.14
	HCV	42 (48.27%)	7 (70%)	0.33
	HIV-HCV	28 (32.18%)	2 (20%)	0.66
Sexual intercourse with both men and women	HIV	0 (0%)	55 (57.89%)	0.36
	HCV	2 (100%)	47 (49.47%)	0.48
	HIV-HCV	0 (0%)	30 (31.57%)	0.85
Unprotected sexual intercourse with men who have sex with men (MSM)	HIV	0 (0%)	55 (57.89%)	0.36
	HCV	2 (100%)	47 (49.47%)	0.48
	HIV-HCV	0 (0%)	30 (31.57%)	0.85
Unprotected sexual intercourse with HIV+ partners	HIV	0 (0%)	43 (53.08%)	0.01
	HCV	3 (100%)	38 (46.91%)	0.01
	HIV-HCV	0 (0%)	22 (27.16%)	0.07
Abstinence	HIV	45 (65.21%)	6 (30%)	0.89
	HCV	41 (59.42%)	7 (35%)	0.07
	HIV-HCV	26 (37.68%)	3 (15%)	0.48
Condom use during the last penetrating sexual intercourse	HIV	11 (55%)	40 (57.97%)	0.1
	HCV	10 (50%)	38 (55.07%)	0.69
	HIV-HCV	6 (30%)	23 (33.33%)	0.01
Sexual intercourse in exchange for drugs/money	HIV	1 (16.66%)	54 (59.34%)	0.1
	HCV	4 (66.66%)	45 (49.45%)	0.69
	HIV-HCV	0 (0%)	30 (32.96%)	0.21
Unprotected sexual intercourse with sex workers	HIV	1 (33.33%)	54 (57.44%)	0.81
	HCV	2 (66.66%)	47 (50%)	1
	HIV-HCV	0 (0%)	30 (31.91%)	0.58
Newly diagnosed sexually transmitted diseases (STDs)	HIV	30 (75%)	25 (43.86%)	<0.05
	HCV	26 (65%)	23 (40.35%)	0.02
	HIV-HCV	17 (42.5%)	13 (22.8%)	0.06
Intravenous drug users (IDUs)	HIV	28 (57.14%)	27 (56.25%)	1
	HCV	36 (73.46%)	13 (27.08%)	<0.05
	HIV-HCV	19 (38.77%)	11 (22.91%)	0.14
Unprotected sexual intercourse with IDUs	HIV	18 (54.54%)	33 (60%)	0.65
	HCV	25 (75.75%)	22 (40%)	<0.05
	HIV-HCV	11 (33.33%)	18 (32.72%)	0.4
Common use of injecting materials (syringes/needles)	HIV	7 (36.84%)	44 (62.85%)	0.11
	HCV	13 (68.42%)	35 (50%)	0.02
	HIV-HCV	3 (15.78%)	26 (37.14%)	0.1
Common use of injecting materials (spoons/filters/water)	HIV	7 (38.88%)	44 (61.97%)	0.19
	HCV	13 (72.22%)	35 (49.29%)	0.01
	HIV-HCV	3 (16.66%)	26 (36.62%)	0.13

Legend: the results of the chi-squared test with the number and percentage of positive tests among exposed and unexposed persons in the inmate population; *p* < 0.05 was considered significant.

**Table 4 viruses-16-01279-t004:** The correlation between risk factors and the outcomes of HIV, HCV and HIV-HCV coinfection screening tests in the general population (logistic regression).

Risk Factor		Logistic Regression							
		OR Reactive vs. Non-Reactive *	*p* Value	OR Age **	*p* Value	OR Exposed vs. Non-Exposed ***	*p* Value	OR Exposed vs. No Response ****	*p* Value
Sexual intercourse with men	HIV	0.04	<0.001	0.98	0.2	2.56	0.02	-	-
	HCV	4	<0.001	0.99	0.92	1.54	0.12	-	-
	HIV-HCV	4	<0.001	0.99	0.92	3.66	0.12	-	-
Sexual intercourse with women	HIV	0.11	<0.001	0.98	0.36	0.29	3	-	-
	HCV	0.01	<0.001	1	0.95	0.31	0.12	-	-
	HIV-HCV	0.01	<0.001	1	0.95	0.31	0.12	-	-
Sexual intercourse with both men and women	HIV	<1	0.98	0.98	0.31	>1	0.98	0.98	1
	HCV	<1	0.99	1	0.98	>1	0.99	1	1
	HIV-HCV	<1	0.99	1	0.98	>1	0.99	1	1
Unprotected sexual intercourse with men who have sex with men (MSM)	HIV	0.48	0.38	0.98	0.44	0.09	<0.001	<1	0.98
	HCV	<1	0.99	1	0.98	>1	0.99	0.99	1
	HIV-HCV	<1	0.99	1	0.98	>1	0.99	0.99	1
Unprotected sexual intercourse with HIV+ partners	HIV	2.06	0.28	0.98	0.36	0.01	<0.001	<1	0.98
	HCV	0.11	0.04	1	0.99	0.06	0.01	<1	0.99
	HIV-HCV	0.11	0.04	1	0.99	0.06	0.01	<1	0.99
Abstinence	HIV	0.08	<0.001	0.98	0.31	0.27	0.02	-	-
	HCV	8	<0.001	1	0.99	1.02	0.81	-	-
	HIV-HCV	8	<0.001	1	0.99	1.02	0.81	-	-
Condom use during the last penetrating sexual intercourse	HIV	0.01	<0.001	0.98	0.2	10.42	2	-	-
	HCV	<1	0.98	0.99	0.92	>1	0.99	-	-
	HIV-HCV	<1	0.98	0.99	0.92	>1	0.99	-	-
Sexual intercourse in exchange for drugs/money	HIV	1.21	0.83	0.98	0.38	0.04	<0.001	-	-
	HCV	0.73	0.69	1	0.91	7	<0.001	-	-
	HIV-HCV	0.73	0.69	1	0.91	7	<0.001	-	-
Unprotected sexual intercourse with sex workers	HIV	1.45	0.66	0.98	0.47	0.03	<0.001	<1	0.98
	HCV	<1	0.99	1	0.98	>1	0.99	1	1
	HIV-HCV	<1	0.00	1	0.98	>1	0.99	1	1
Newly diagnosed sexually transmitted diseases (STDs)	HIV	0.29	0.19	0.98	0.27	0.16	0.02	7.05	0.03
	HCV	<1	0.99	1	0.98	>1	0.99	>1	0.99
	HIV-HCV	<1	0.99	1	0.98	>1	0.99	>1	0.99
Intravenous drug users (IDUs)	HIV	0.97	0.97	0.98	0.45	0.03	<0.001	<1	0.98
	HCV	1	0.04	1.17	0.06	<1	0.92	<1	1
	HIV-HCV	1	0.04	1.17	0.06	<1	0.92	<1	1
Unprotected sexual intercourse with IDUs	HIV	0.81	0.8	0.98	0.32	0.06	<0.001	<1	0.98
	HCV	0.43	0.19	1	0.93	9	<0.001	<1	0.99
	HIV-HCV	0.43	0.19	1	0.93	9	<0.001	<1	0.99
Common use of injecting materials (syringes/needles)	HIV	1.94	0.44	0.98	0.3	0.02	<0.001	<1	0.98
	HCV	0.69	0.56	1	0.9	0.92	<0.001	>1	0.99
	HIV-HCV	0.69	0.56	1	0.9	4	<0.001	<1	0.99
Common use of injecting materials (spoons/filters/water)	HIV	0.4	0.2	0.98	0.26	0.11	<0.001	<1	0.98
	HCV	0.15	0.12	1	0.91	<1	0.99	<1	0.99
	HIV-HCV	0.15	0.12	1	0.91	<1	0.99	<1	0.99

Legend: the results of the logistic regression in the general population. * The likelihood of a reactive test compared to a non-reactive one among individuals exposed to the risk factor. ** The likelihood of age influencing the screening test outcome in individuals from the same group (with or without a risk factor). *** The likelihood of a positive test among individuals exposed to the risk factor compared to those who were not exposed. **** The likelihood of a positive test among individuals exposed to the risk factor compared to those who did not recall exposure to the risk factor; *p* < 0.05 was considered significant.

**Table 5 viruses-16-01279-t005:** The correlation between risk factors and the outcomes of HIV, HCV and HIV-HCV coinfection screening tests in the inmate population (logistic regression).

Risk Factor		Logistic Regression							
		OR Reactive vs. Non-Reactive *	*p* Value	OR Age **	*p* Value	OR Exposed vs. Non-Exposed ***	*p* Value	OR Exposed vs. No Response ****	*p* Value
Sexual intercourse with men	HIV	0.36	0.45	0.96	0.17	18.57	<0.05	-	-
	HCV	<1	0.99	0.95	0.06	>1	0.98	-	-
	HIV-HCV	<1	0.99	0.95	0.06	>1	0.98	-	-
Sexual intercourse with women	HIV	3.65	0.19	0.97	0.35	0.26	0.06	-	-
	HCV	2.25	0.44	0.96	0.13	0.45	0.34	-	-
	HIV-HCV	2.25	0.44	0.96	0.13	0.45	0.34	-	-
Sexual intercourse with both men and women	HIV	<1	0.98	0.98	0.44	>1	0.98	-	-
	HCV	<1	0.99	0.96	0.15	>1	0.99	-	-
	HIV-HCV	<1	0.99	0.96	0.15	>1	0.99	-	-
Unprotected sexual intercourse with men who have sex with men (MSM)	HIV	<1	0.98	0.98	0.44	>1	0.98	-	-
	HCV	<1	0.99	0.96	0.15	>1	0.99	-	-
	HIV-HCV	<1	0.99	0.96	0.15	>1	0.99	-	-
Unprotected sexual intercourse with HIV+ partners	HIV	<1	0.99	0.98	0.56	>1	0.99	>1	0.98
	HCV	<1	0.99	0.96	0.2	>1	0.99	>1	0.99
	HIV-HCV	<1	0.99	0.96	0.2	>1	0.99	>1	0.99
Abstinence	HIV	2.59	0.34	0.99	0.73	0.23	9	0.52	0.38
	HCV	2.68	0.37	0.96	0.17	0.31	92	0.19	0.14
	HIV-HCV	2.68	0.37	0.96	0.17	0.31	0.09	0.19	0.14
Condom use during the last penetrating sexual intercourse	HIV	2.36	0.41	0.98	0.48	1.12	0.82	0.76	0.75
	HCV	2.37	0.46	0.95	0.11	1.17	0.77	0.26	0.26
	HIV-HCV	2.37	0.46	0.95	0.11	1.17	0.77	0.26	0.26
Sexual intercourse in exchange for drugs/money	HIV	0.47	0.58	0.97	0.28	8.87	0.05	-	-
	HCV	<1	0.99	0.95	0.08	>1	0.99	-	-
	HIV-HCV	<1	0.99	0.95	0.08	>1	0.99	-	-
Unprotected sexual intercourse with sex workers	HIV	0.91	0.95	0.98	0.43	3.13	0.36	-	-
	HCV	<1	0.99	0.96	0.11	>1	0.99	-	-
	HIV-HCV	<1	0.99	0.96	0.11	>1	0.99	-	-
Newly diagnosed sexually transmitted diseases (STDs)	HIV	2.77	0.3	1.002	0.93	0.25	4	-	-
	HCV	1.95	0.52	0.97	0.33	0.44	0.07	-	-
	HIV-HCV	1.95	0.52	0.97	0.33	0.44	0.07	-	-
Intravenous drug users (IDUs)	HIV	2.04	0.36	0.98	0.51	1.03	0.94	-	-
	HCV	1.86	0.55	0.97	0.28	0.52	0.15	-	-
	HIV-HCV	1.86	0.55	0.97	0.28	0.52	0.15	-	-
Unprotected sexual intercourse with IDUs	HIV	2.63	0.32	0.98	0.38	1.36	0.49	0.65	0.58
	HCV	2.65	0.37	0.95	0.1	1.15	0.76	0.22	0.19
	HIV-HCV	2.65	0.37	0.95	0.1	1.15	0.76	0.22	0.19
Common use of injecting materials (syringes/needles)	HIV	1.62	0.63	0.97	0.25	3.32	0.03	1.71	0.53
	HCV	1.38	0.78	0.94	51	4.05	0.04	0.64	0.73
	HIV-HCV	1.38	0.78	0.94	51	5.04	0.04	0.64	0.73
Common use of injecting materials (spoons/filters/water)	HIV	1.62	0.63	0.97	0.29	2.86	0.05	1.56	0.6
	HCV	1.37	0.79	0.95	0.06	3.06	0.06	0.6	0.69
	HIV-HCV	1.37	0.79	0.95	0.06	3.06	0.06	0.6	0.69

Legend: the results of the logistic regression in the inmate population. * The likelihood of a reactive test compared to a non-reactive one among individuals exposed to the risk factor. ** The likelihood of age influencing the screening test outcome in individuals from the same group (with or without a risk factor). *** The likelihood of a positive test among individuals exposed to the risk factor compared to those who were not exposed. **** The likelihood of a positive test among individuals exposed to the risk factor compared to those who did not recall exposure to the risk factor; *p* < 0.05 was considered significant.

## Data Availability

Research data are available on request.

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
