# Peer review of "HIV, HCV and HIV-HCV Coinfections in the General Population versus Inmates from Romania"

_viruses, 2024, doi:10.3390/v16081279_

Round 1
Reviewer 1 Report
Comments and Suggestions for Authors
the study looks at the general population in romania, 806, 64% women
and prison inmates 97, 93 percent male
how were these patients selected?
would the statistics on seroprevalence been different if 97 percent of the general population would have been selected to be male?
it would be good to clarify the design of the study, and the comparison between the prison population and the general population used
Author Response
Thank you for your suggestions, which helped us improve the quality of the manuscript. We made the requested changes in the manuscript in red, and our point-by-point responses are as follows:
- “The study looks at the general population in romania, 806, 64% women and prison inmates 97, 93 percent male how were these patients selected?”
The participants were randomly chosen; all the people from both populations who accepted to be tested and be part of the project, were included in the study and tested free of charge for both HIV and HCV infections.
We acknowledge the fact that the percentages of males and females in the two populations are disproportionate, however, the National Administration of Penitentiaries’ report from 31 January 2023, stated that there were 22,030 men (95.62%) and 1,010 women (4.38%) imprisoned in Romania in incarceration units from their administration. Given this high proportion of imprisoned men, the percentage 92.78% of men is representative for the inmates’ population.
As you have suggested, information about this issue was included in the manuscript (in red) in the Materials and Methods section (lines 79 – 81 and 111-114) and in the Discussions section (lines 400 – 406), as well as a new paragraph covering recent information from the National Administration of Penitentiaries’ report ([25]) to support our data.
- “Would the statistics on seroprevalence been different if 97 percent of the general population would have been selected to be male?”
This is a very good point, we plan to address this issue in future studies by increasing the sample size, which will rise the significance of the statistical analysis and help us validate our findings about HIV and HCV risk factors in both populations.
- “It would be good to clarify the design of the study, and the comparison between the prison population and the general population used.”
As stated before in this response, the participants were randomly chosen and a table containing the general characteristics of the patients was added in the Results section (lines 132 – 137).
Reviewer 2 Report
Comments and Suggestions for Authors
In this study, authors performed an observational, prospective study on 903 patients to analyze the epidemiological links of HIV, HCV and HIV-HCV coinfections. They compared the risk factors in incarcerated people with the ones from general population in Romania. It is a very meaningful topic.
Major points:
1. The major concern is that the small number of enrolled individuals. Do authors consider that it may limit the meaning of this study?
Minor points:
1. Full names of abbreviations are suggested to be provided when first appeared. For example, HIV and HCV in the Abstract.
2. It is suggested that the general characteristics of the patients can be presented as a table.
3. Proofreading is suggested. Some typos are found in the manuscript.
Author Response
- “In this study, authors performed an observational, prospective study on 903 patients to analyze the epidemiological links of HIV, HCV and HIV-HCV coinfections. They compared the risk factors in incarcerated people with the ones from general population in Romania. It is a very meaningful topic.”
Thank you very much for your appreciation and all your suggestions, which helped us improve our manuscript. We made the requested changes in the manuscript in red, and our point-by-point responses are as follows:
- “Major points:
- The major concern is that the small number of enrolled individuals. Do authors consider that it may limit the meaning of this study?”
This is a very good point, we have acknowledged this issue in the manuscript however, to our knowledge, this is the first study to compare the risk factors in incarcerated people with the ones from general population in Romania bringing up new pieces of information in this domain of research in our country. Moreover, we added a short paragraph at the end of the Materials and Methods section, in order to emphasize the statistical relevance of the present results (lines 112 – 114).
We also plan to address this subject in future studies by increasing the sample size, which will rise the significance of the statistical analysis. This will help us validate our findings and provide more definitive conclusions about HIV and HCV risk factors in both populations.
- “Minor points:
- Full names of abbreviations are suggested to be provided when first appeared. For example, HIV and HCV in the Abstract.”
Thank you very much for taking time to carefully review the text, we added the full names of abbreviations, as you have suggested.
2. “It is suggested that the general characteristics of the patients can be presented as a table.”
We have listed the general characteristics of the patients as a table and added it in the Results section (lines 132 – 137).
3. “Proofreading is suggested. Some typos are found in the manuscript.”
As suggested, we re-checked the text for typos and made the necessary changes.
Round 2
Reviewer 1 Report
Comments and Suggestions for Authors
they have made requested changes and should be accepted for publication
Author Response
Thank you for your kind remarks and suggestions.